# A mathematical modelling framework for the regulation of intra-cellular OCT4 in human pluripotent stem cells

L. E. Wadkin[1]*, S. Orozco-Fuentes[2], I. Neganova[3], M. Lako[4], N. G. Parker[1], A. Shukurov[1]

**1** School of Mathematics, Statistics and Physics, Newcastle University, Newcastle upon Tyne, United Kingdom, **2** Department of Mathematics, Physics and Electrical Engineering, Northumbria University, Newcastle upon Tyne, United Kingdom, **3** Institute of Cytology, RAS St Petersburg, Novosibirsk, Russia, **4** Bioscience Institute, Newcastle University, Newcastle upon Tyne, United Kingdom

* laura.wadkin@newcastle.ac.uk

**Data Availability Statement:** All relevant data and coding files will be made available from the Newcastle University data repository https://data.

## Abstract

Human pluripotent stem cells (hPSCs) have the potential to differentiate into all cell types, a property known as pluripotency. A deeper understanding of how pluripotency is regulated is required to assist in controlling pluripotency and differentiation trajectories experimentally. Mathematical modelling provides a non-invasive tool through which to explore, characterise and replicate the regulation of pluripotency and the consequences on cell fate. Here we use experimental data of the expression of the pluripotency transcription factor OCT4 in a growing hPSC colony to develop and evaluate mathematical models for temporal pluripotency regulation. We consider fractional Brownian motion and the stochastic logistic equation and explore the effects of both additive and multiplicative noise. We illustrate the use of time-dependent carrying capacities and the introduction of Allee effects to the stochastic logistic equation to describe cell differentiation. We conclude both methods adequately capture the decline in OCT4 upon differentiation, but the Allee effect model has the advantage of allowing differentiation to occur stochastically in a sub-set of cells. This mathematical framework for describing intra-cellular OCT4 regulation can be extended to other transcription factors and developed into predictive models.

## Introduction

Human pluripotent stem cells, hPSCs, have the ability to self-renew through repeated divisions and to differentiate into a wide range of cell types, a property known as pluripotency. The pluripotency of hPSCs is their defining characteristic, central to their applications regenerative medicine [1, 2]. However, hPSCs exhibit complex behaviour and the *in-vitro* control of their differentiation trajectories is challenging.

Pluripotency is controlled by an inter-regulatory network of pluripotency transciption factors, PTFs, including the genes OCT4, SOX2 and NANOG [3–5]. The destabilisation of PTFs and their interaction with chemical signalling pathways result in differentiation away from the

ncl.ac.uk/articles/software/ModellingOCT4_data/
14474412.

**Funding:** LEW would like to acknowledge support
from the London Mathematical Society (Early
Career Fellowship ECF-1920-106). ML
acknowledges the Biotechnology and Biological
Sciences Research Council for grant BB/I020209/1.
IN acknowledges the grant from the Russian
Government 641 Program for the recruitment of
the leading scientists into 641 Russian Institution
of Higher Education 14.w03.31.0029 and Russian
Science Foundation grant number 21-75-20132.
The funders had no role in study design, data
collection and analysis, decision to publish, or
preparation of the manuscript.

**Competing interests:** The authors have declared
that no competing interests exist.

pluripotent state and into a specialised cell [3, 6, 7]. This decision of a cell to either remain pluripotent or to differentiate is known as its fate decision. It is unknown how much cell fate decisions are led by inherited factors, as opposed to environmental factors and intra-cellular signalling as even clonal (genetically identical) cells under apparently identical conditions make different fate decisions [8]. In many *in-vitro* experiments the differentiation of hPSC populations is induced and facilitated by a differentiation agent, such as BMP4 [9, 10].

A narrow range of PTF expression is necessary to maintain cell pluripotency, with both high and low expressions causing a shift from the pluripotent state [11, 12] and even small fluctuations can bias cell fate decisions [13]. Furthermore, the PTFs are inherited asymmetrically as a cell divides, biasing the fate of the daughter cells and contributing to colony heterogeneity [14–16] with the decision to differentiate largely determined before any differentiation stimulus is introduced [14]. Given the likely large number of factors involved in the fate decisions and our limited knowledge of their nature, the probabilistic framework to modelling PTF dynamics appears to be the most suitable. However, careful, experiment-based quantification of the stochastic, temporal dynamics of PTFs is necessary to examine the resulting effects on cell fate.

Statistical analysis and mathematical modelling are deepening our understanding of hPSC behaviours and guiding the development of experimental protocols [17]. Recent mathematical models of cell pluripotency focus on describing the network of PTFs and the resulting cell fate decisions to guide the optimisation and control of pluripotency *in-vitro* [17–19]. These models are informed by recent studies of fluctuations of PTFs throughout colonies [13, 14, 20] and the spatial patterning of differentiation [21, 22]. Many models use coupled differential equations based on the Hill equations [23] describing changes in concentrations of molecules to describe PTF fluctuations [24–26]. Others use network analysis frameworks [27] or explore the mechanical aspects of the cell behaviour when both the model and data are complex [28]. These models often aim to describe the whole PTF regulatory network and it can be difficult to estimate the model parameters accurately from experimental data [26].

Here we focus on the methodology of building such mathematical models using experimental data for the transcription factor OCT4. Although the OCT4 dynamics will be affected by many external factors and the remainder of the PTF network, there are benefits to considering each PTF in isolation as the crucial first step; firstly, this simplifies the model development process, allowing each element to be explored in a systematic way and secondly, the results provide a basis for comparison to the other PTFs (e.g., NANOG and SOX2) from similar experiments. Similarly, although interesting spatial patterning effects are seen in OCT4 [29], we will consider only the intra-cellular OCT4 behaviour through time. These simpler models can be used to describe the stochastic nature of PTF regulation on shorter time scales and explore the effects of each PTF on cell fate, before their development into coupled models of the entire pluripotency regulatory network.

Here we systematically explore various mathematical models for the temporal regulation of the PTF OCT4. We aim to identify the optimal set of mathematical tools required to reproduce the key quantitative features of experimental observations from Ref. [14] and the additional quantitative analysis of this dataset from Ref. [29]. The framework discussed can be applied in future to other experimental datasets. Since PTF fluctuation is inherently stochastic [14, 20, 30, 31], we focus on different forms of well-established stochastic models to describe the behaviour, namely: fractional Brownian motion and the stochastic logistic equation. The aim is to describe the PTFs as microstates before considering the macrostate of cellular pluripotency. Firstly, we introduce the experimental data and outline the key features of OCT4 to be described mathematically. Next, we explore fractional Brownian motion and the stochastic logistic equation for simulating temporal OCT4 before any cell differentiation occurs. We

consider different types of random noise (additive and multiplicative [32, 33]) and their effects. Finally, we examine the use of shifting carrying capacities and Allee effects to simulate a reduction in OCT4 towards the differentiated state.

## Experimental OCT4 fluctuations

We use experimental data of OCT4 expression in a growing hESC colony from Ref. [14] and our previous analysis of this data in Ref. [29] to guide model development. Although focused on one experiment, the mathematical framework outlined here is easily adaptable to other experimental results. We use the experimental analysis in Ref. [14] and Ref. [29] to illustrate the applicability of such models to PTF regulation. Here we summarise the experiment and main features of the data to be described by a mathematical model.

### Experiment summary

This experiment was carried out by Purvis Lab (University of North Carolina, School of Medicine), and is published in Ref. [14]. The OCT4 levels (mean OCT4-mCherry fluorescence intensity) in a human embryonic stem cell colony were determined and cells were live-imaged for 68 hours. The colony begins from 30 cells and grows over 68 hours (817 time frames) to 463 cells, with 1274 cell cycles elapsing within this time. After 43 hours, the hESCs were treated with (100 ng/ml) bone-morphogenetic protein 4 (BMP4) to induce their differentiation towards distinct cell fates. The cell IDs and ancestries were extracted along with their OCT4 immuno-fluorescence intensity values (reported in arbitrary fluorescence units, a.f.u.). The measurements of the OCT4 signal at 5 minute intervals, results in a set of evenly sampled discrete observations for each cell, OCT4$(t_0)$, OCT4$(t_1)$, . . ., OCT4$(t_n)$, where $t_0$ is the time of cell 'birth' and $t_n$ the time of cell division. The values of $t_n$ range from 0.25–30 hours across the population, with a mean ± standard deviation of 10.3 ± 4hours.

To classify the cells as either self-renewing (pluripotent) or differentiated, the mean nuclear OCT4 and CDX2 were quantified at 68 hours. A two-component mixed Gaussian distribution representing pluripotent (OCT4+/CDX2−) and differentiated (OCT4−/CDX2+) categories was fit to the data, with hESCs assigned to each group if >99% confidence was met. Cells not reaching the confidence threshold were allocated the 'unknown' category. Further details are presented in Ref. [14]. Using these fates, the cell population was traced back in time, spanning multiple cell divisions, with each earlier cell labelled according to this pro-fate. The colony begins from 14 pluripotent, 2 differentiated and 14 'unknown' category cells. In this paper we consider only the pluripotent and differentiated fate groups. Note that for times pre-BMP4 (before 43 hours), the fate classification is a pro-fate based on the fate of the cells descendants.

### Temporal OCT4 features

The OCT4 expression of (pro-)pluripotent and (pro-)differentiated cells for the whole experimental time (68 hours) is shown in Fig 1(a). At 43 hours the differentiation agent BMP4 is added, after which there is a decline in OCT4 expression in the (pro-)differentiated cells. The (pro-)pluripotent cells retain their OCT4 expression levels. The distribution of all OCT4 expressions pre-differentiation is shown in Fig 1(b), with temporal distributions in Fig 1(c) and 1(d) for pluripotent and differentiated pro-fate cells respectively. A detailed analysis of the experimental data is provided in Ref. [29]. For simplicity, and due to the distinct behavioural differences identified pre- and post-differentiation, we first consider modelling the temporal behaviour pre-BMP4 before moving on to the effect of cell differentiation. From the

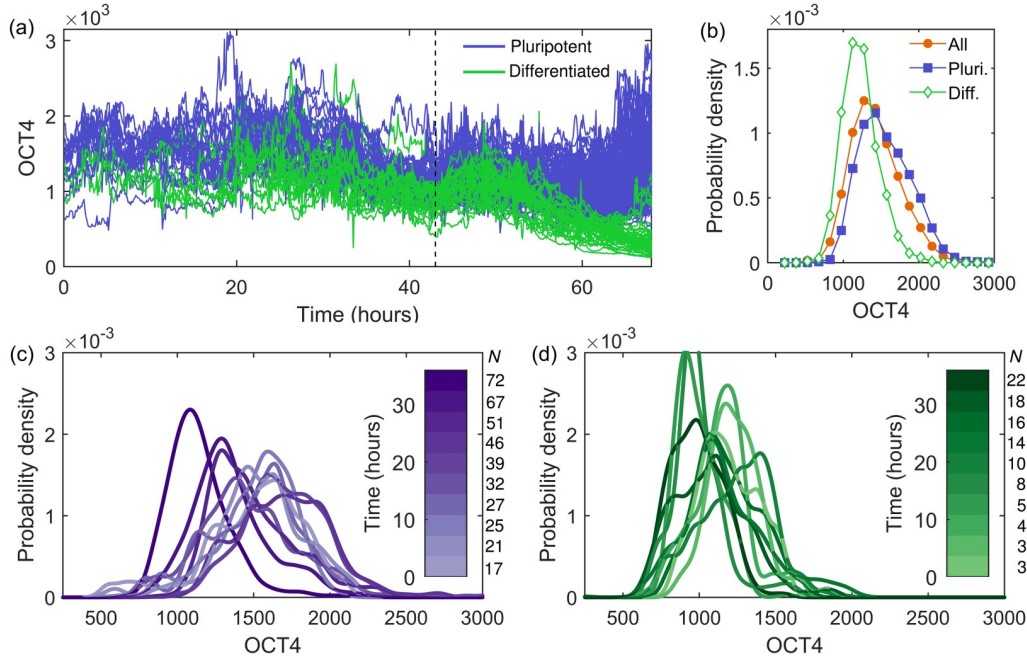

**Fig 1. Experimental OCT4 properties.** (a) The temporal OCT4 expression for all (pro-)pluripotent (purple) and (pro-) differentiated (green) cells up to 68 hours. At 43 hours (vertical dashed line) the differentiation agent BMP4 was added. Pre-differentiation: (b) The distribution of all OCT4 expressions for all (orange circles), pro-pluripotent (purple squares) and pro-differentiated (green diamonds) cells. The distribution of OCT4 expression for binned time intervals between zero and 43 hours for (c) pro-pluripotent and (d) pro-differentiated cells. The colour bar shows the time of the bin centre. The numbers of cells included in each bin are indicated by *N* to the right of the colour bar.

experimental data and analysis in Ref. [29], we identify several key features (labelled F1–6 with F1–4 pre-differentiation features and F5–6 post-differentiation features) to capture in model development, as follows:

**Pre-differentiation.**

F1. The time series exhibit stochastic noise, shown in Fig 1(a), with a mean Hurst exponent of 0.38±0.09 in both (pro-)pluripotent and (pro-)differentiated cells, calculated in Ref. [29]. A Hurst exponent <0.5 indicates anti-persistence in the time series, with increases in OCT4 more likely to be followed by decreases, and vice versa. Further details on the Hurst exponent are given in the (S1 File), along with the distribution of all calculated Hurst exponents for every cell (with >50 time frames available) and the distribution of their standard deviations in S1 Fig in S1 File.

F2. Pro-differentiated cells show reduced OCT4 expression throughout, shown in Fig 1(a) and 1(b).

F3. The distribution of all OCT4 expressions from (pro-)pluripotent cells is positively skewed, resulting from a reduction in expression at later times, shown in Fig 1(b) and 1(c).

F4. The distribution of all OCT4 expressions from (pro-)pluripotent cells show a temporal shift in the mode, with a reduction in expression with time, shown in Fig 1(c). The distributions are statistically different, confirmed by the Kolmogorov-Smirnov test at the 95% level.

**Post-differentiation.**

F5. At the end of the experiment differentiated cells are classified according to their OCT4 and CDX2 expressions. These differentiated cells show a pronounced reduction in OCT4 upon BMP4 addition (43 hours), as shown in Fig 1(a).

F6. There is a clear and natural separation between the two classified groups post-BMP4 based on their OCT4 levels, with differentiated cells showing reduced OCT4 and pluripotent cells retaining OCT4 expression, as shown in Fig 1(a).

In the next section we explore mathematical models to identify which can capture one, some, or all, of these key behavioural features. We aim to descriptively reproduce the features for this particular experiment, but note that future work will focus on which of these properties are inherent for all hPSCs and modelling the behavioural properties of OCT4 more globally.

## Results

### Modelling OCT4 pre-differentiation

In the following sections we systematically explore the use of different stochastic models as a framework for temporal OCT4 regulation, aiming to capture the experimental behaviour described in features F1–6 above and shown in Fig 1. The model development process allows the identification of the key mathematical tools and important biological parameters required to descriptively reproduce the data. All the models discussed in the following sections have the same basis, with the initial conditions and cellular division incorporated using the algorithmic base model detailed below.

**Base model.**

1. We begin with a chosen initial number of cells, $N = N_0$, to match the experimental conditions.

2. Each of the $N$ cells are allocated an initial OCT4 value. This is extracted probabilistically from the kernel density fitting to the experimental distribution of initial OCT4, OCT4 ($t = 0$), shown in Fig 2(a) and S2(a) Fig in S1 File.

3. Each of the $N$ cells are allocated a cell cycle duration. This is extracted probabilistically from the kernel density fitting to the experimental distribution of cell cycle times for all pre-BMP4 cells, shown in Fig 2(b) and S2(b) Fig in S1 File. Each cell's starting position in its cell cycle is chosen uniformly.

4. For each of the $N$ cells the OCT4 values for the duration of their cell cycle are simulated using one of the stochastic models.

5. Each of the $N$ cells divide into two cells at the end of their cell cycle. For each of the two daughter cells, their initial OCT4 value is set to the pre-division OCT4 value of the mother cell.

6. Repeat steps 4 and 5 for a specified number of division (mitosis) events. Note that as each OCT4 series is generated for a whole cell life time, the number of division events sets the end point for the model, rather than timesteps. We use the number of division events required to ensure all divisions occurring prior to the time point of interest have occurred, e.g., 600 events for pluripotent cells and 200 for differentiate cells comfortably exceeds the 68 hour full experimental time. For a shorter time window of interest, the number of

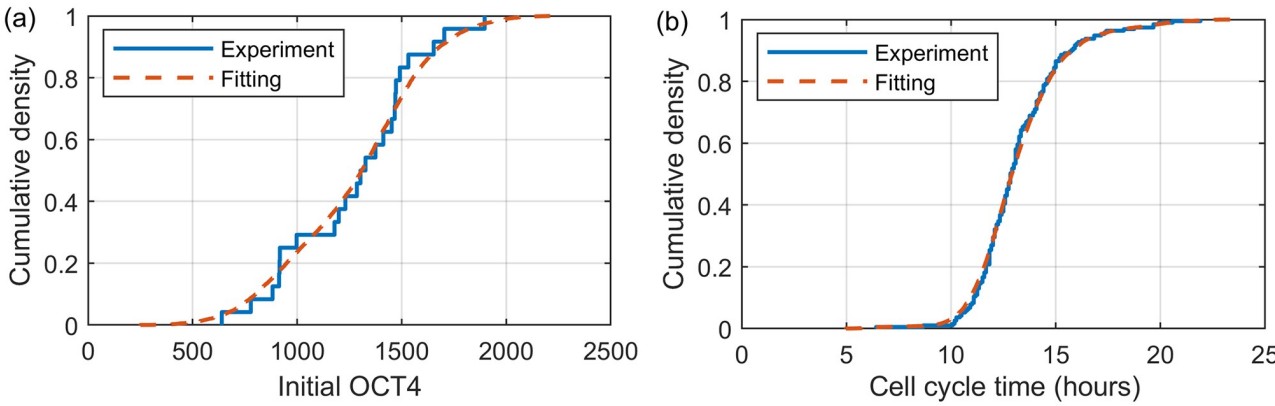

**Fig 2. The initial conditions used in the common base model.** (a) The cumulative density function of the experimental initial OCT4 values (blue), OCT4($t = 0$), with kernel density fitting (orange dashed). (b) The cumulative density function of the experimental cell cycle duration times (blue) for all cells pre-BMP4 addition with kernel density fitting (orange dashed).

division events can be reduced, or we can retain excess division events and remove time frames outside the window of interest post-simulation.

When the cell cycle times are generated in step 3 it is necessary to specify how much of the cell cycle has already elapsed. If all cells begin at the start of their cell cycle at the start of the simulation then divisions will be synchronised, shown in S2(c) Fig in S1 File. This synchronisation can be avoided by starting cells at different points in their cell cycles, as shown in S2(d) Fig in S1 File. We do not know exactly how the cell cycles are aligned in the cells in this experiment and so the resulting colony growth could lie between the synchronised and asynchronous examples. Here we choose to continue with the asynchronous cell cycles. Note that the cell cycle distribution post-BMP4, shown in S2(b) Fig in S1 File, shows a decrease in cell cycle times, but for simplicity, and since the colony growth does not affect the method of OCT4 generation, we keep the same distribution throughout.

Although here we have used the analysis of the experimental data to inform the initial conditions and the cell cycle simulation, this is flexible and can easily be adapted to other experimental results. The OCT4 regulation itself is captured in step 4 and is open to many mathematical modelling techniques. In the next section we use the experimental results from Ref. [14, 29] to systematically build a stochastic model using fractional Brownian motion and the stochastic logistic equation.

**Anti-persistent OCT4 fluctuations.** One possibility for a simple model of OCT4 fluctuation is to assume that the expression fluctuates symmetrically with no preferred trends or correlations. Mathematically this would be descried by a Wiener process, analogous to the physical phenomenon of Brownian motion in one dimension and the starting point for many random walk models. However, the analysis of experimental OCT4 expression described above and in Ref. [29] has shown that the OCT4 evolution is anti-persistent, with an average Hurst exponent of $H = 0.38$ (feature F1). This signifies that increases in OCT4 are more likely to be followed by decreases, and vice versa. The Hurst exponent $H \neq 0.5$ indicates that the fluctuations in OCT4 cannot be captured by simple Brownian motion.

Instead we consider the generalisation, fractional Brownian motion (fBm). Unlike Brownian motion, fBm allows for non-independent increments and hence persistence or anti-persistence. An fBM random function of time $t$, $B_H(t)$, with an initial value $B_H(0)$ and time

increments $B_H(t - s)$ is defined by

$$B_H(t) = B_H(0) + \frac{1}{\Gamma(H + 0.5)} \int_{-\infty}^{0} \left[ (t - s)^{H-0.5} - (-s)^{H-0.5} \right] dB(s)$$
$$+ \frac{1}{\Gamma(H + 0.5)} \int_{0}^{t} (t - s)^{H-0.5} dB(s),$$

(1)

where $H$ is the Hurst exponent and $\Gamma$ is the gamma function [34]. There are several ways to simulate fBm, either exact or approximate [35–37]. Here we use the Matlab function *ffgn* [38] which uses the circulant embedding technique for $H < 0.5$ [39] and Lowen's method [40] for $H > 0.5$ (both exact methods) to simulate the fractional Brownian noise. There is also an inbuilt Matlab function *wfbm* (available in the Wavelet toolbox) which uses a wavelet based approximate simulation method [41].

We can use fBm to simulate OCT4 over time (step 4 of the base model) with a scaling parameter $\sigma$ which controls the level of noise, i.e., $\sigma B_H$. Example realisations of the fractional noise, corresponding fBm functions, and simulated OCT4 for varying $H$ are shown in S3 Fig in S1 File to illustrate the effect of the Hurst exponent. The parameter $\sigma$ is estimated from the experimental data (for all pre-BMP4 cells) as the standard deviation of $\Delta OCT4 = OCT4(t) - OCT4(t - 1)$, leading to $\sigma \approx 90$. Each time series for OCT4 can then be generated as $OCT4(t = 0) + \sigma B_H$.

For simplicity, we first consider both cell fates together with $N = 16$ cells, made up of 14 pro-pluripotent and two pro-differentiated cells to correspond to the experimental data [14]. For cells in the experimental colony $H = 0.38$ [29]. A comparable simulation using fBm with 16 initial cells, $H = 0.38$, and $\sigma = 90$ is shown in S4(a) and S4(b) in S1 File. Note that although we simulate from a limited number of starting cells, the number of OCT4 values generated over 40 hours due to the 5 minute increments and cellular division is approximately 30000. It is clear from S4(a) and S4(b) in S1 File that this level of anti-persistent regulation from the Hurst exponent is not sufficient to keep the OCT4 expression within the range seen in the experiment.

A common mathematical method of limiting variables is to impose boundary conditions, either absorbing or reflecting. In this case, absorbing boundary conditions suggest that once the OCT4 level reaches either the upper or lower boundary, the cell is theoretically removed in some way from the experiment and its OCT4 time series does not continue. There is no indication or biological evidence of particularly high or low OCT4 expressions resulting in cell death experimentally [14, 29]. However, high or low OCT4 expressions do accompany cell differentiation [13], so the removal of cells via the boundary condition could correspond to the differentiation of cells if we were to consider modelling pluripotent cells only. We can estimate the lower boundary to be equal to zero to correspond to the positive nature of the OCT4 measurements. The upper boundary is more difficult to define; here we take 2500 (as 99.9% of the data points fall below this value) for illustrative purposes. The OCT4 simulation for fBm with absorbing boundary conditions is shown in S4(c) and S4(d) in S1 File.

The introduction of reflecting boundary conditions results in the OCT4 expressions being reflected back in the opposite direction upon reaching the set boundary. Biologically this corresponds to an additional regulatory effect which could be internal to the cell, i.e., if the OCT4 level in a cell becomes too low, there is systematic regulation to increase it (and vice versa). The simulation using fBm with reflecting boundary conditions (again at 0 and 2500) is shown in S4(e) and S4(f) in S1 File. Reflecting boundary conditions produce a result more similar to the experiment than absorbing boundary conditions since cells are not artificially removed, but it still creates a sharper distribution boundary than seen experimentally. Additionally,

although the boundary conditions somewhat artificially force the OCT4 into the desired range, the spread of the overall expressions is not well captured.

This illustrates that the anti-persistence from the Hurst exponent alone is not sufficient to capture the OCT4 regulation seen in the experiment, even with boundary conditions. The imposition of any boundary conditions would also require further investigation to elucidate their nature and the biological implications. Particularly for the upper boundary, further work would be needed to constrain its value. For this reason, we next choose to investigate other methods of introducing regulatory effects. We can still incorporate fBm noise into other models to generate the anti-persistence seen experimentally and capture feature F1. In the next section we consider describing temporal OCT4 with the stochastic logistic equation and explore the regulatory effects of a limiting carrying capacity.

**The stochastic logistic equation.**   In this section we explore the application of the stochastic logistic equation (SLE) to simulating temporal OCT4 regulation. The logistic equation is a widely used model of population dynamics characterized by the growth rate of the population, encapsulated by the parameter $r$, and its optimal size called the carrying capacity, denoted $K$. We adapt the logistic equation to the experimental data available, using the model for OCT4 variation, rather than the traditional population size. Since fBm alone does not fully capture the regulatory behaviour of OCT4, some additional effects are clearly important. We consider the SLE with additive noise, multiplicative noise, and the effect of a time-dependent carrying capacity. For simplicity, we again consider the two cell fates together initially.

There are several ways stochasticity can be introduced into the logistic equation, e.g., additive noise, multiplicative noise, a noisy parameter $r$ or carrying capacity $K$. Both additive and multiplicative noise can be used to regulate gene expression [32]. The most straightforward of these is additive noise which can be introduced by adding a noise term to the net rate of change in the PTF. This noise does not depend on the system dynamics of OCT4 and therefore can represent constant sources of external noise, or constant noise within measurements. Additive noise can also result from molecular fluctuations within chemical reactions [33, 42]. The SLE with additive random scatter to describe OCT4, $O$, over time, $t$, is then

$$\frac{dO}{dt} = rO\left(1 - \frac{O}{K}\right) + \sigma_A \xi, \tag{2}$$

where $\xi$ is the stochastic noise (e.g., Wiener/Brownian noise, or fBM noise) and $\sigma_A$ is a scaling parameter controlling the magnitude of the scatter.

We can use the experimental data (pre-BMP4) to estimate and constrain some of the parameters that appear in Eq (2). In keeping with the anti-persistence, the noise $\xi$ corresponds to fBm noise with the Hurst exponent $H = 0.38$ and the scaling parameter is again the standard deviation of $\Delta$OCT4, $\sigma_A = 90$. We can also estimate the carrying capacity as the median of all the experimental OCT4 values, $K = 1290$. This leaves the parameter $r$, which controls the growth rate of OCT4 from the initial conditions to the carrying capacity. Once OCT4 is fluctuating around the carrying capacity, $r$ has the effect of controlling the strength of the regulation to the carrying capacity value, in opposition with the stochastic fluctuations. Throughout our models we estimate $r$ to give an appropriate qualitative fit to the experimental data. The OCT4 dynamics simulated using Eq (2) with $r = 0.02$ is illustrated in Fig 3(a) and 3(b). Although the regulatory effect of the carrying capacity works well to capture the upper bound of OCT4 expression, an additional boundary condition at small values of OCT4 is still required (if the stochasticity gives rise to $O < 0$ then $dO/dt < 0$ resulting in $O \rightarrow -\infty$). A distinguishing feature not captured by the model is the positive skew in the distribution of all occurring OCT4 values, shown in Fig 1(b) and overlaid in Fig 3(b). The model promotes tighter regulation above the

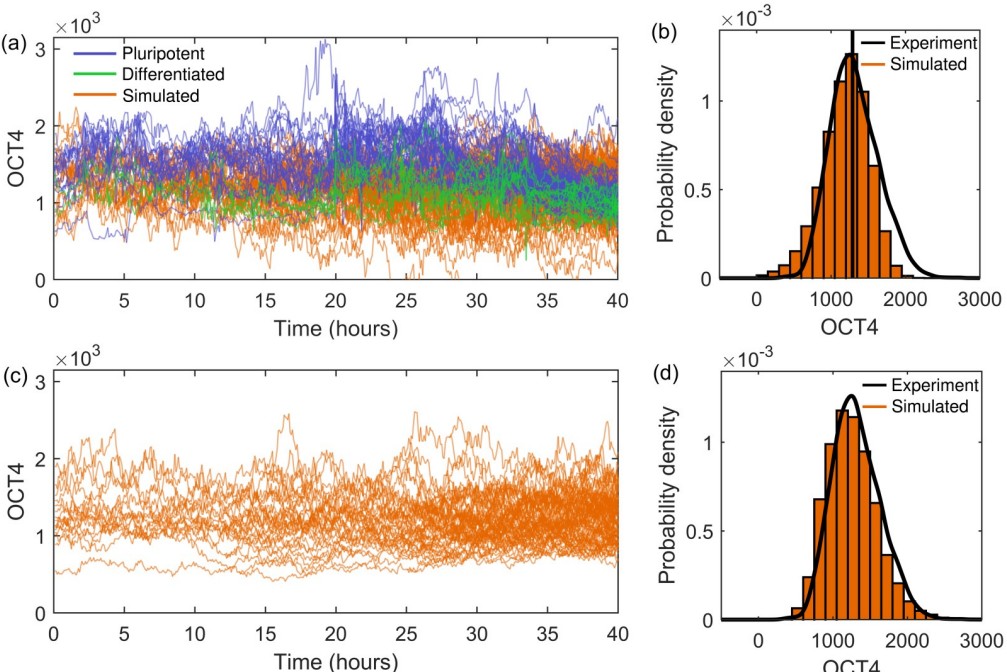

**Fig 3. Comparison of experimental and simulated OCT4 using the SLE with either additive or multiplicative noise.** (a) Simulated OCT4 expression (orange) using the SLE with additive noise, Eq (2), with 16 initial cells, $r = 0.02$, $K = 1290$, $\sigma_A = 90$ and fBM noise with $H = 0.38$, with an absorbing boundary condition at zero. The experimental OCT4 is shown in purple and green for pluripotent and differentiated cells, respectively. (b) The corresponding histogram of simulated OCT4 expression using Eq (2) with the experimental distribution and estimated carrying capacity (vertical line, $K = 1290$) in black. (c) Simulated OCT4 expression using the SLE with multiplicative noise, Eq (3), with 16 initial cells, $r = 0.005$, $K = 1290$, $\sigma_M = 0.0045$ and fBM noise with $H = 0.38$. (d) The corresponding histogram of simulated OCT4 expression with the experimental distribution in black.

carrying capacity than below it, resulting in few OCT4 expressions above the carrying capacity. However, in the experimental OCT4, we do see large fluctuations at high OCT4 values (corresponding to values above the carrying capacity). This suggests that the stochasticity (the magnitude of the fluctuations) has some dependence on the current state of the system (the current value of OCT4).

Whereas the additive noise in Eq (2) has no dependence on the state of the system and corresponds to making $dO/dt$ symmetrically noisy, multiplicative noise changes depending on the current conditions, i.e., the current value of OCT4, and originates from fluctuations in cellular components that indirectly cause variation in transcription factor dynamics [33, 42]. In the case of our temporal OCT4 simulation, multiplicative noise can be used to generate a scatter in the simulated data which has a greater magnitude when the system is close to the carrying capacity (thus resulting in more stochastically high OCT4 expressions) and a reduced magnitude when far away from the carrying capacity. Hints of this behaviour can be seen in Fig 1(a), with larger fluctuations apparent in the cells exhibiting above average OCT4 expression. For simulating the SLE with multiplicative noise we first consider the rearrangement of the logistic equation,

$$\frac{d\ln(O)}{dt} = r\left(1 - \frac{O}{K}\right).$$

Applying the substitution $X = \ln(O)$ and adding stochasticity $\xi$ with noise scaling parameter $\sigma$ gives

$$\frac{dX}{dt} = r\left(1 - \frac{e^X}{K}\right) + \sigma_{\mathrm{M}}\xi, \qquad (3)$$

which can then be used to simulate $X = \ln(O)$, with the dynamics of OCT4 recovered from $O = e^X$. Example realisations of Eq (3) for both $X$ and $O$ are shown in S5 and in S1 File to illustrate the effect of multiplicative noise in a typical logistic growth scenario for varying $\sigma$. The result is amplified noise for stochasticity occurring above the carrying capacity.

The temporal OCT4 dynamics simulated using the SLE with multiplicative noise, Eq (3), with fBM noise with $H = 0.38$, $K = 1290$ and free parameters $r = 0.005$ and $\sigma_{\mathrm{M}} = 0.0045$ (chosen for illustrative purposes) for 16 initial cells are shown in Fig 3(c). The multiplicative noise results in cells with expressions above the carrying capacity exhibiting increased stochasticity, with lower expression cells showing tighter regulation. The simulated distribution has a slight positive skew and is qualitatively similar to the experimental distribution, as shown in Fig 3(d).

This model provides a good basis for capturing the experimental results across the whole time period and is an improvement on the SLE with additive noise. However, it does not take into account the different cell fates (feature F2), and the evolving temporal positive skew (feature F3) in the pluripotent cell group, shown in Fig 1(c). In the following sections we consider the two cell fates separately and discuss two methods of including the temporal skew in the pluripotent cell group: the SLE with a transition between dominant additive and dominant multiplicative noise, and the SLE with a time-dependent carrying capacity.

**SLE with noise transition.** Firstly, to capture the changing temporal skew for pluripotent cells (feature F3), we could include both additive and multiplicative noise because different noise types reflect different aspects in the cell behaviour [32, 33] and both appear to be involved in the experimentally observed evolution of OCT4. If additive noise is dominant at early times, and multiplicative noise at later times, the resulting OCT4 distribution will be symmetric at early times and skewed at later times. The increasing dominance of intrinsic transcription noise would require further investigation as to its biological implications. We can consider the following rearrangement of the stochastic logistic equation with additive noise

$$\frac{d\ln(O)}{dt} = r\left(1 - \frac{O}{K}\right) + \frac{\sigma_{\mathrm{A}}}{O}\xi_1,$$

make the substitution $X = \ln(O)$ and introduce the multiplicative noise term $\sigma_{\mathrm{M}}\xi_2$,

$$\frac{dX}{dt} = r\left(1 - \frac{e^X}{K}\right) + \frac{\sigma_{\mathrm{A}}}{e^X}\xi_1 + \sigma_{\mathrm{M}}\xi_2. \qquad (4)$$

As before, we can simulate the dynamics for $X$ and recover the dynamics for $O = e^X$.

For simplicity, we can consider the change between additive and multiplicative noise as a switch for pluripotent cells with additive noise only for $0 < t < 20$h and multiplicative noise only for $t \geq 20$h. The switch time is chosen as the time at which the distribution of OCT4 becomes positively skewed in the experimental data, shown in Fig 1(c). The parameters are specified in Table 1. Since differentiated cells show reduced OCT4 expression throughout (feature F2), they are given a lower carrying capacity. The results for the OCT4 dynamics within this regime are shown in Fig 4. The reduced carrying capacity for differentiated cells results in their lower expression throughout, shown in Fig 4(a). The overall OCT4 expression distributions in Fig 4(b) are well described. The temporal distributions in Fig 4(c) illustrate the effect

**Table 1. Fitting parameters for the OCT4 expression for pluripotent and differentiated cells using the SLE with both multiplicative and additive noise, Eq (4).** At 20 hours the noise switches from additive to multiplicative noise in the pluripotent cells. * indicates a free parameter, with the remaining parameters constrained by the experimental data.

|  | Parameter | $t < 20$h | $t \geq 20$h |
|---|---|---|---|
| Pluripotent | $r^*$, (5 min)$^{-1}$ | 0.01 | |
|  | $K$, a.f.u. | 1290 | |
|  | $\sigma_A$ | 90 | 0 |
|  | $\sigma_M{}^*$ | 0 | 0.05 |
| Differentiated | $r^*$, (5 min)$^{-1}$ | 0.01 | |
|  | $K$, a.f.u. | 1000 | |
|  | $\sigma_A$ | 90 | |
|  | $\sigma_M{}^*$ | 0 | |

of the noise switch in the pluripotent cells, with the appearance of a positive skew at later times, while the expression of differentiated cells in Fig 4(d) remains symmetrical at later times, descriptively capturing features F1, F2 and F3. 6.

Although this model captures the overall distribution and provides the desired temporal change in skew (which could be further smoothed with a more sophisticated time-dependent noise function, feature F3), it does not result in a shift in the mode expression as drastic as the one apparent in Fig 1(c) (feature F4). For this we consider implementing a time-dependent carrying capacity in the next section.

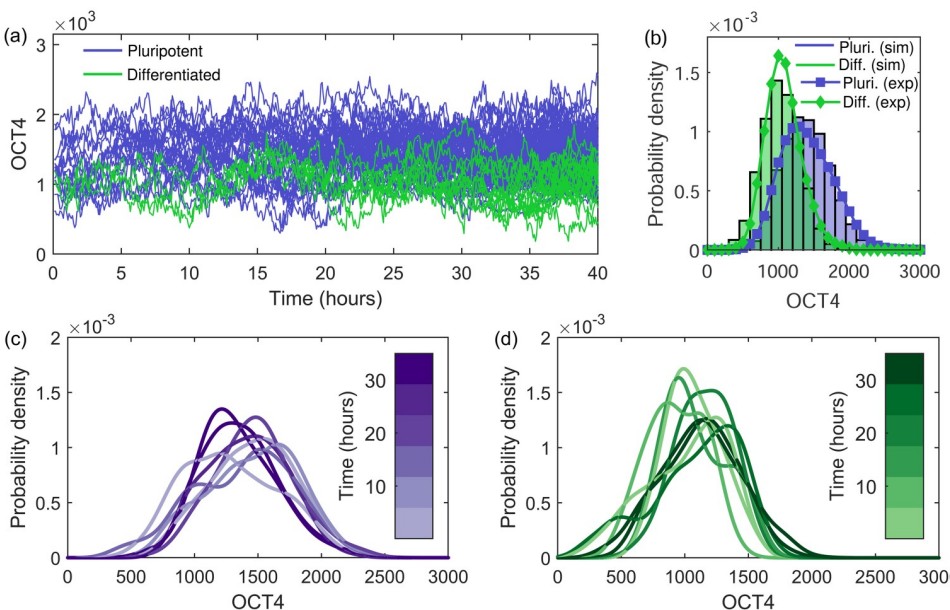

**Fig 4. The dynamics of OCT4 simulated using the SLE with a switch between additive and multiplicative noise.** (a) The OCT4 dynamics between zero and 40 hours for 14 pro-pluripotent (purple) and two pro-differentiated (green) initial cells following the SLE with both additive and multiplicative noise, Eq (4), with the parameters specified in Table 1. For pro-pluripotent cells the noise changes from additive to multiplicative at 20 hours. (b) The distribution of all simulated OCT4 values for pro-pluripotent (purple) and pro-differentiated (green) cells with the corresponding experimental distributions overlaid. The temporal distributions for (c) pro-pluripotent and (d) pro-differentiated cells split by time intervals.

**Table 2. Fitting parameters for generating OCT4 expression for pro-pluripotent and pro-differentiated cells using the SLE with additive and multiplicative noise, and a time-dependent carrying capacity, Eq (5).** * indicates a free parameter, with the remaining parameters constrained by the experimental data.

|  | Parameter | $t < 25h$ | $t \geq 25h$ |
|---|---|---|---|
| Pluripotent | $r^*$, $(5\ \text{min})^{-1}$ | 0.015 | |
|  | $K$, a.f.u. | 1500 | 1000 |
|  | $\sigma_A{}^*$ | 30 | |
|  | $\sigma_M{}^*$ | 0.035 | |
| Differentiated | $r^*$, $(5\ \text{min})^{-1}$ | 0.015 | |
|  | $K$, a.f.u. | 1100 | 1000 |
|  | $\sigma_A{}^*$ | 20 | |
|  | $\sigma_M{}^*$ | 0.03 | |

**SLE with time-dependent carrying capacity.**  To reproduce the significant shift in the mode for the pluripotent cells, shown in Fig 1(c) (feature F4), we can employ a time-dependent carrying capacity. We use the stochastic logistic equation for all cells, with both multiplicative and additive noise, as in Eq (4), and a carrying capacity which varies with time,

$$\frac{dX}{dt} = r\left(1 - \frac{e^X}{K(t)}\right) + \frac{\sigma_A}{e^X}\xi_1 + \sigma_M\xi_2. \tag{5}$$

For simplicity, we will consider one change of carrying capacity at 25 hours, as at this time the reduction in the average OCT4 begins. We can estimate the carrying capacity as the median OCT4 between zero and 25 hours resulting in $K_p \approx 1500$ and $K_p \approx 1100$ for pluripotent and differentiated cells, respectively. Post-25 hours, the carrying capacities can be estimated as $K \equiv K_p K_d \approx 1000$. This reduction in the carrying capacity will initiate the corresponding reduction in the mode of the distribution over time we see experimentally. The OCT4 dynamics using time-dependent carrying capacities in Eq (5) for 14 pro-pluripotent and two pro-differentiated cells, with the model parameters summarised in Table 2, are shown in Fig 5.

The lower carrying capacity results in consistently lower OCT4 expression for the differentiated cells (feature F2), as shown in Fig 5(a) and 5(b). The overall distribution of OCT4 expressions is well described, shown in Fig 5(b). The model captures the shift to lower OCT4 values in pluripotent cells (feature F4), shown in the temporal distribution in Fig 5(c). The time-dependent carrying capacity function $K(t)$ could be further developed to represent a smooth temporal transition and can be adapted to capture other significant increases and decreases in expression. The noise parameter choices could also be refined to additionally capture the change in the temporal skew using time-dependent multiplicative noise.

Here we have outlined some possible techniques for simulating temporal OCT4 using the SLE with different modes of fBm stochasticity and a time-dependent carrying capacity. The fBm stochasticity allows the recovery of the Hurst exponent in all models (feature F1), with two separate cell populations allowing for flexibility in capturing the systematic lower OCT4 in the pro-differentiated cells (feature F2). Multiplicative noise can introduce a skew in the overall distribution of OCT4 values (as we see in pro-pluripotent cells, feature F3) and a time-dependent carrying capacity can reproduce reductions in OCT4 with time (as we see in pro-pluripotent cells pre-BMP4, feature F4). Note that we aim to illustrate the application of such a model and describe a framework which could be used to capture some of the global properties of experimental data sets. Further work is now required to elucidate the appropriate parameter choices with further experiments and explore their biological implications.

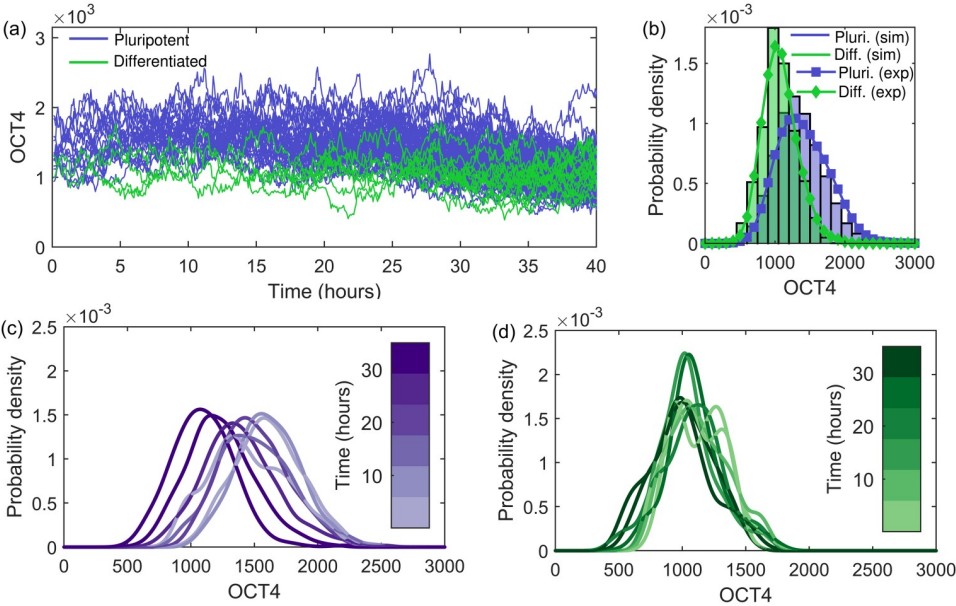

**Fig 5. The dynamics of OCT4 simulated using the SLE with a time-dependent carrying capacity.** (a) The OCT4 dynamics between zero and 40 hours for 14 pro-pluripotent (purple) and two pro-differentiated (green) initial cells following the SLE with both additive and multiplicative noise and a time-dependent carrying capacity, Eq (5), with the parameters specified in Table 2. For pro-pluripotent cells there is a large reduction in carrying capacity at 25 hours, causing a visible decline in OCT4 after this time. (b) The distribution of all simulated OCT4 values for pro-pluripotent (purple) and pro-differentiated (green) cells with the corresponding experimental distributions overlaid. The temporal distributions for (c) pro-pluripotent and (d) pro-differentiated cells split by time intervals.

## Simulating cell differentiation

In the previous section we considered modelling temporal OCT4 regulation before any differentiation stimulus (BMP4) is added, corresponding to the time interval $0 < t < 43h$ in the experimental colony [14, 29]. The addition of BMP4 causes a significant reduction in OCT4 expression in the differentiated cells (feature F5) shown in Fig 1(a). The mean OCT4, shown in Fig 6(a) also shows the clear reduction in differentiated cells. For completeness, the median and mode experimental OCT4 are shown in S6(a) and S6(b) in S1 File. We explore two methods of modelling this reduction in OCT4 as differentiation is induced. Firstly, we apply the SLE with a time-dependent carrying capacity as discussed previously, and secondly, we consider the use of the SLE with an Allee effect. Although not seen in this experiment, it should be noted that high OCT4 values can also correspond to cell differentiation [13].

**Differentiation with a time-dependent carrying capacity.** We previously employed the SLE with a time-dependent carrying capacity, Eq (5), to simulate a moderate reduction in the average OCT4 expression post-25 hours, as shown in Fig 5. We could extend this technique to simulate the more drastic reduction in OCT4 seen when the differentiation stimulus is added.

As in the previous section, we can estimate the carrying capacities for $t < 25h$ as $K_p \approx 1500$ and $K_p \approx 1100$ for pluripotent and differentiated cells, respectively. For $t > 25h$ we can simulate the reduction in OCT4 (particularly pronounced in the pluripotent cells) with a reduction of the carrying capacity to $K_p = K_d \approx 1000$. For the differentiated cells, a reduction to $K_d \approx 300$ in the time interval $t > 43h$ corresponds to cell differentiation. These shifting carrying capacities, along with the other model parameters are given in Table 3. The dynamics under this

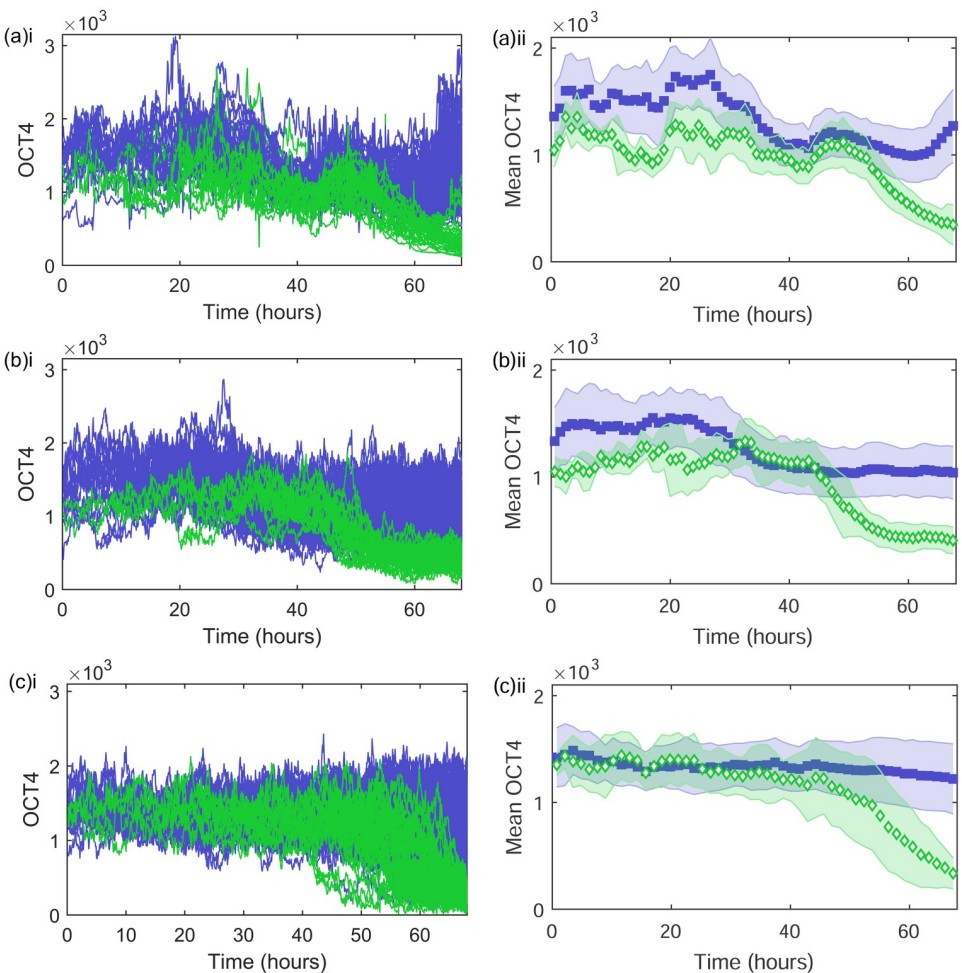

**Fig 6. The experimental and simulated dynamics of OCT4 upon differentiation at 43 hours.** The (a) experimental (i) OCT4 and (ii) mean OCT4 with time. The (b) simulated (i) OCT4 and (ii) mean OCT4 with time with differentiation induced at 43 hours using a time-dependent carrying capacity, Eq (5), with the parameters specified in Table 3. The (c) simulated (i) OCT4 and (ii) mean OCT4 wth time with differentiation induced at 43 hours by introducing an Allee effect term to the SLE, Eq (7), with $r$ = 0.025, $K$ = 1290, $\sigma_A$ = 35, $\sigma_A$ = 0.035 and $A$ = 1000.

**Table 3. Fitting parameters for the OCT4 expression of pluripotent and differentiated cells using the SLE with additive and multiplicative noise, and a time-dependent carrying capacity, Eq (5), to capture induced differentiation.** * indicates a free parameter, with the remaining parameters constrained by the experimental data.

| | Parameter | $0 \leq t < 25$h | $25 \leq t < 43$h | $43 \leq t < 68$h |
|---|---|---|---|---|
| Pluripotent | $r^*$, (5 min)$^{-1}$ | 1500 | 0.015 | 1000 |
| | $K$, a.f.u. | | 1000 | |
| | $\sigma_A^*$ | | 30 | |
| | $\sigma_M^*$ | | 0.035 | |
| Differentiated | $r^*$, (5 min)$^{-1}$ | 0.015 | 0.015 | 0.008 |
| | $K$, a.f.u. | 1100 | 1000 | 300 |
| | $\sigma_A^*$ | | 20 | |
| | $\sigma_M^*$ | | 0.03 | |

regime are shown in Fig 6(b) and S6(c) and S6(d) in S1 File. The time-dependent carrying capacity leads to the reduction of OCT4 in the differentiated cell group, well capturing the dynamics of (features F4 and F5).

This model could be further refined by the use of a more sophisticated function for the time-dependent carrying capacity, which could be estimated from experimental data such as that in Ref. [13, 14]. The model could also easily be adapted to include a population of cells exhibiting high OCT4 values pre-differentiation, with a corresponding increase in their carrying capacity. However, the model would remain purely descriptive, with pro-pluripotent and pro-differentiated cells defined from the outset with different behavioural rules. Next we consider using the SLE with an Allee effect to simulate differentiation and identify the different cell fate types.

**Differentiation with an Allee effect.** Another possible method of modelling induced differentiation is the SLE with a demographic Allee effect. Allee effects are traditionally used for modelling population numbers, with the effect inhibiting population growth at low densities as observed in both animal and cell populations [43–45]. The deterministic logistic equation for OCT4 expression $O$ with this effect incorporated has the form

$$\frac{dO}{dt} = rN\left(1 - \frac{O}{K}\right)\left(\frac{O - A}{K}\right),$$ (6)

where $A$ is critical point at which the Allee effect occurs. Note that there are other methods of simulating Allee effects through e.g., difference equations [46, 47] and Lotka-Voltera models [48, 49]. Here we use the logistic equation for consistency with our previous modelling results.

The effect of the Allee term in Eq (6) on both $dO/dt$ and the OCT4 expression $O$ for an example system is illustrated in S7 Fig in S1 File. For a weak Allee effect, $A < O(t = 0)$, the rate of change $dO/dt$ remains positive for $O < K$ but is significantly suppressed. For a stonger Allee effect, $A > O(t = 0)$, $dO/dt$ is negative for $O < K$ and results in the OCT4 expression declining to zero. It is this declining effect we can employ to simulate the reduction in OCT4 expression for the differentiated cells. The Allee effect can be introduced at a certain time point resulting in either continued suppressed growth or a decline to zero. Examples of 'switching on' both weak and strong Allee effects during logistic growth are shown in S8 Fig in S1 File.

For simulating OCT4 expression through the differentiation process with the SLE, we can switch on the Allee effect term at the time the differentiation agent is added (43 h). If the OCT4 expression is below $A$, then the Allee effect will be strong and the OCT4 will decline to zero. The stochasticity in the system will mean that only some of the cells will meet this condition, with others having an OCT4 expression greater than $A$, and therefore continuing with (suppressed) logistic growth. The stochasticity will also result in this effect taking place at all times past 43h, so the differentiation process will happen at different times for different cells. The SLE for $X = \ln(O)$ with additive fBm noise $\xi_1$ and multiplicative fBm noise $\xi_2$ is

$$\frac{dX}{dt} = r\left(1 - \frac{e^X}{K}\right)\left(\frac{e^X - A}{K}\right) + \frac{\sigma_A}{e^X}\xi_1 + \sigma_M\xi_2,$$ (7)

where $A$ is the Allee effect critical point.

The OCT4 dynamics for 16 cells simulated with the SLE, Eq (4), for $t < 43$h and the SLE with an Allee effect, Eq (7), for $t \geq 43$h with $r = 0.025$, $K = 1290$, $\sigma_A = 35$, $\sigma_M = 0.035$ and $A = 1000$ are shown in Fig 6(c) and S6(e) and S6(f) Fig in S1 File. Here the fates of each cell are identified at the end of the simulation, with the cells whose OCT4 has reduced as a result of the Allee effect classed as differentiated, and the cells whose OCT4 has remained constant as pluripotent. The model captures the reduction of OCT4 in the differentiated subset of cells

whilst keeping a remaining pluripotent cell population (features F4 and F5). However, the OCT4 in the pro-differentiated group pre-Allee effect is no lower than for the pro-pluripotent cell group, unlike in the experimental results (feature F2). Furthermore, an additional model would be required to introduce differentiated cells with high OCT4 values.

## Discussion

We have explored different modelling techniques for describing temporal OCT4 regulation, guided by previous analysis of experimental OCT4 expression in a growing hESC colony [14, 29], particularly fractional Brownian motion and the stochastic logistic equation. A differentiation agent, BMP4, was added to the cells at 43 hours and results in the reduction of OCT4 expression in the differentiated cells. Although not seen here, it is also possible for high OCT4 expression to accompany cell differentiation [13]. Pre-BMP4 we identified the key features (F1–4) including an anti-persistent stochasticity, and for pluripotent cells a temporal skew and shifting mode in the distribution of all OCT4 expressions. All the models discussed follow a common base model which sets up the initial conditions and describes cell proliferation. We then focus on different mathematical methods of generating the temporal OCT4 expressions for the cell population within this base model. The simulated populations consist of 16 cells (with 14 pro-pluripotent and two pro-differentiated) resulting in approximately 30000 simulated OCT4 expressions. We have taken a systematic approach, gradually building complexity to illustrate the methodology of developing stochastic models for biological systems.

Firstly, we consider modelling the OCT4 dynamics pre-BMP4, i.e., for $t < 43$hours. The analysis in Ref. [29] revealed that OCT4 values fluctuate stochastically with anti-persistence and a Hurst exponent, $H$, of 0.38 (feature F1), suggesting the use of fractional Brownian motion (fBm) [34]. There is also further experimental evidence that gene expressions and transcription factor dynamics display fractal characteristics [50]. The Hurst exponent for genetic expression in *E. Coli* has been found to be ≈0.8, showing long-range memory with persistence [50]. It is thought that these stochastic fractal dynamics can lead to phenotypic diversity [50, 51]. Another study in a variety of bacteria found ranges of $H$ between 0.3–0.8 for different genes, showing a negative correlation with the gene length [52]. The use of fBm is particularly common in financial modelling [53–55], but it has also been used to describe diffusion within crowded fluids (such as the cytoplasm of cells) [56] and the kinetics of transcription factors [57]. The stochasticity from fBm results in a wider range of OCT4 values at later times than seen experimentally (an effect which is exacerbated with time).

The range of OCT4 can be controlled artificially with boundary conditions (either absorbing or reflecting), but the overall distribution of all OCT4 values is not well captured. It is also unclear whether these boundary conditions are biologically appropriate as OCT4 expression is regulated by a complex range of factors across the transcriptional, post-transcriptional and epigenetic regulation levels [3, 7, 58, 59]. Interestingly, mechanical limits to transcription have been shown to naturally generate bounds to transcriptional noise [60]. A boundary condition at zero corresponds to the fact that OCT4 expression never becomes negative with the upper boundary representing a maximum possible value of arbitrary fluorescence intensity. This also raises the question of the biological implications of the removal of cells through absorbing boundaries or the recovery of expression through reflecting boundaries? One possibility for absorbing boundaries for pro-pluripotent cells is to represent differentiation happening at both the upper and lower boundary [13]. Although fBm alone is not sufficient to capture the experimental behaviour, it does (by design) capture the anti-persistence ($H$ = 0.38) and so in all later model iterations we use fBm noise to generate the stochasticity.

A somewhat less artificial method of keeping the OCT4 values within range is to use the stochastic logistic equation (SLE), which has a regulating parameter of the carrying capacity, *K*, which represents the maximum amount of OCT4 that can be expressed within each individual cell. Note that this could be due to limits on the expression of OCT4 due to other members of the regulatory network which cause its down-regulation. In our model, the stochasticity allows for some fluctuations above *K*. Similarly to the boundary conditions this maximum value depends on the complex inter-regulatory network of OCT4, however, we estimate the value of the carrying capacity from the experimental results as the median of all OCT4 values (taking into account the stochasticity allowing for $O > K$).

There are many possible sources of noise within the system and various ways to simulate stochastic series [61]. Noise inherent in molecular fluctuations results from stochastic chemical reactions (e.g., noise in the rate constants) and emerges as additive noise as it is independent of the variables of the system [33, 42]. There is also multiplicative noise originating from fluctuations in other cellular components that indirectly cause variation in transcription factor dynamics [33, 42]. We consider both additive and multiplicative noise, shown in Fig 3. The introduction of multiplicative noise creates larger fluctuations above the carrying capacity, qualitatively similar to those seen in the experiment. This results in a distribution of all OCT4 values well matched to the experiment, with the slight positive skew being captured. Both additive and multiplicative noise can be used to regulate gene expression, with multiplicative noise allowing small deviations in transcription rates to lead to large fluctuations in protein productions [32]. Future work could compare these results to simulation with a Gillespie algorithm to draw links with the rates of reactions involved in the OCT4 regulation for intrinsic noise [62], with further extensions for external noise [63].

A property not captured by the SLE with either additive or multiplicative noise is the time-dependency of this positive skew (feature F3). It occurs only at later times, and only in pluripotent cells, shown in the time-discretised distributions of OCT4 in Fig 1(c). This temporal skew can be captured by the SLE with both additive and multiplicative noise, with the type of noise time-dependent; additive noise at early times produces symmetrical distributions of OCT4, with multiplicative noise at later times producing skewed distributions, shown in Fig 4. Here we changed the noise function stepwise, but this could be further smoothed using a more sophisticated time-dependent noise function. The biological implications of a change in dominance in noise types would be an interesting avenue for future work. This could be linked to experimental results which show that when OCT4 production is high (at early stages in the cell cycle [13]) the system does not take into account the current levels of OCT4 in the cell [64], leading to additive noise more predominantly in the earlier stages of the cell cycle.

Another interesting property of the experimental OCT4 is the decline in expression for pluripotent cells post-25 hours (feature F4) shown in Fig 1(c). We consider capturing this behaviour using the SLE with a time-dependent carrying capacity. Since this parameter is likely to depend on a large number of biological factors, it is not unreasonable to expect that it may change with environmental conditions and experimental time. We consider the pluripotent and differentiated cells separately, each with a different carrying capacity, corresponding to the suggestion that the decision to differentiate is determined pre-differentiation stimulus [14]. The carrying capacity for both cell groups is reduced at 25 hours, resulting in a decline in OCT4 expression, particularly for the pluripotent cell group with originally higher expression. Although this technique well describes the experimental results (shown in Fig 5), it requires multiple parameters which need to be elucidated from further experimental data.

We then consider modelling the OCT4 regulation for all times, including the decline in expression due to the addition of the differentiation stimulus. We extend the time-dependent carrying capacity approach, reducing the carrying capacity further for the differentiated cell

group at 43 hours. This well captures the decline in OCT4 upon differentiation (feature F5), along with the more subtle decline in pluripotent cells, shown in Fig 6(b). Here we have used a stepwise change in the parameter $K$, but this is easily adjustable to other experimental results and more sophisticated functions could be used to capture other trends. Similarly, a population of high OCT4 differentiated cells could be introduced with a corresponding increase in their carrying capacity. The pro-differentiated cells are identified from the outset and although this is not biologically unreasonable, with evidence that cell fate is determined pre-differentiation agent [14], the model itself does not produce the two fate groups (feature F6) which limits its future capacity to develop into a predictive model. It is also worth noting that although the time-dependency of $K$ increases the model flexibility to capture trends, it also increases the number of parameters required to be estimated from the experimental data.

A method of inducing differentiation which naturally produces the two fate groups is the SLE with an Allee effect. Allee effects are well used across mathematical biology [43–45], but we are not aware of their application to pluripotency transcription factor expression. The Allee effect results in a decline to zero for cells whose OCT4 expression fluctuates below the critical point $A$. The stochasticity in the system means that this condition is met for only some of the cells, causing the formation of a differentiated cell group with reducing or zero OCT4 and a pluripotent cell group with stable OCT4 expression at the carrying capacity, shown in Fig 6(c). This model is limited to describing low OCT4 differentiated cells as seen in this experiment and high OCT4 differentiation would need to be incorporated through another technique. This model could be combined with a time-dependent carrying capacity to capture the decline in expression in pluripotent cells.

A summary of which mathematical models can be used to capture each of the key experimental features is given in Table 4. Since the two differentiation models have distinct advantages, depending on whether it is more appropriate to define differentiated cells from the outset (shifting carrying capacity model) or they occur stochastically (Allee effect model), we have not quantitatively compared the two models. In the future development of these models to predictive models a formal comparison (such as Bayesian Information Criterion) could be applied to aid model selection.

The models discussed here are of a purely descriptive nature, but outline a possible framework for modelling the regulation of OCT4. We have explored systematically a wide range of effects that might be able to reproduce rather fine details in the experimentally observed dynamics of the OCT4 expression and identified an adequate and optimal combination of such effects. However, the resulting model may not be unique and other approaches may be

**Table 4. A summary of the key features identified experimentally and the models used to describe each behaviour.**

|  | Key features | Model |
|---|---|---|
| Pre-differentiation | F1. Stochastic noise with Hurst exponent of 0.38. | Any model using fractional Brownian noise, Eq (1). |
|  | F2. Pro-differentiated cells show reduced OCT4 throughout. | Incorporated through initial conditions by considering two populations in any version of the SLE. |
|  | F3. Positive skew of all pro-pluripotent OCT4 expressions. | SLE with multiplicative noise, Eqs 3 and 4. |
|  | F4. Reduction in pro-pluripotent OCT4 post 25 hours. | SLE with time-dependent carrying capacity, $K(t)$, Eq (5). |
| Post-differentiation | F5. Reduction in OCT4 expression for some cells. | SLE with either time-dependent carrying capacity $K(t)$ or Allee effect, Eq (5) or Eq (7). |
|  | F6. Separation into pluripotent and differentiated groups. | SLE with time-dependent carrying capacity $K(t)$ or Allee effect, Eq (5) or Eq (7), but only with the Allee effect does this happen stochastically. |

viable. To justify any model of this kind and to develop it into a prognostic tool for *in-silico* experimentation, it should be assessed and compared with targeted experiments. With this caveat, we believe that the model developed can be used as a provisional prognostic tool and basis for further mathematical model development.

A key next step would be to investigate the effects of the current free parameters through a parameter scan, and to further constrain the free parameters through inference of other experimental data, such as the studies in Refs [13, 65–67]. Another interesting avenue of research would be to compare the models to stochastic network models which consider the wider PTF network [24, 68]. In these models a global trend of anti-persistence is inbuilt through feedback regulations. Here we use the carrying capacity and the persistent noise in the logistic equation to represent the regulatory property of OCT4, without specifying how this regulation occurs. Thus, this could represent negative-feedback systems with other PTFs, with the advantage of being able to consider a single PTF in isolation and encapsulate the regulation with less parameters.

Further time-lapse experiments monitoring single cell PTF expression through colony growth will be useful in confirming which of these properties are inherent to OCT4 expression, and how they vary depending on experimental conditions, and to provide more extensive benchmarking for the modelling approaches and assumptions. It will be informative to apply the same quantitative framework to the other predominant transcription factors, SOX2 and NANOG. Their individual regulatory dynamics could then be compared using the key descriptive parameters, and any systematic differences identified. This information will help build the picture of the wider PTF system with the dynamics of the PTFs considered as part of an inter-linked network. In general, this highlights the need for further temporal experimental data on PTF regulation to build upon this mathematical framework and develop more sophisticated predictive models. These models of the microstate of PTF regulation will help inform longer time-scale models of the pluripotent macrostate.

## Supporting information

**S1 File. Supplementary file containing S1 Appendix and S1–S8 Figs.**
(PDF)

## Acknowledgments

We thank Dr Andrew Golightly for discussions on statistical approaches relevant to this work.

## Author Contributions

**Conceptualization:** N. G. Parker, A. Shukurov.

**Formal analysis:** L. E. Wadkin.

**Investigation:** L. E. Wadkin.

**Methodology:** L. E. Wadkin, S. Orozco-Fuentes, N. G. Parker, A. Shukurov.

**Supervision:** N. G. Parker, A. Shukurov.

**Writing – original draft:** L. E. Wadkin.

**Writing – review & editing:** L. E. Wadkin, S. Orozco-Fuentes, I. Neganova, M. Lako, N. G. Parker, A. Shukurov.

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
