## [Decision Letter · Decision Letter 0]

16 Mar 2021

PONE-D-21-04839

A mathematical modelling framework for the regulation of intra-cellular OCT4 in human pluripotent stem cells

PLOS ONE

Dear Dr. Wadkin,

Thank you for submitting your manuscript to PLOS ONE. The paper was sent to two reviewers, whose reports are included below. While one of the reviewers (reviewer 2) has a positive opinion of the work, reviewer 1 poses major concerns and recommends against publication of the paper in its present form. If you're willing to perform the required major revisions needed, the revised manuscript would be sent again to the same two reviewers, and to a third one for further comments.

We look forward to receiving your revised manuscript.

Kind regards,

Jordi Garcia-Ojalvo

Academic Editor

PLOS ONE

Journal Requirements:

2. Our internal editors have looked over your manuscript and determined that it may be within the scope of our Modeling Cell Proliferation and the Cell Microenvironment Call for Papers. This collection of papers is headed by a team of Guest Editors for PLOS ONE: Aurélie Carlier (Maastricht University), Ravi Iyengar (Ichan School of Medicine at Mount Sinai), Padmini Rangamani (UC San Diego) and Vivek Shenoy (University of Pennsylvania). The Collection will encompass a diverse and interdisciplinary set of research articles modelling cell growth and the cell microenvironment. Additional information can be found on our announcement page: https://collections.plos.org/s/cell-modelling. If you would like your manuscript to be considered for this collection, please let us know in your cover letter and we will ensure that your paper is treated as if you were responding to this call. If you would prefer to remove your manuscript from collection consideration, please specify this in the cover letter.

Reviewers' comments:

Reviewer's Responses to Questions

**Comments to the Author**

1. Is the manuscript technically sound, and do the data support the conclusions?

Reviewer #1: Partly

Reviewer #2: Yes

2. Has the statistical analysis been performed appropriately and rigorously? 

Reviewer #1: No

Reviewer #2: Yes

3. Have the authors made all data underlying the findings in their manuscript fully available?

Reviewer #1: No

Reviewer #2: No

4. Is the manuscript presented in an intelligible fashion and written in standard English?

Reviewer #1: Yes

Reviewer #2: Yes

5. Review Comments to the Author

Reviewer #1: SUMMARY

Wadkin et al. take an already published 68h long experimental dataset of the OCT4 expression (from Wolff et al., 2018) and aim to mathematically model the exact OCT4 expression in each cell of this lineage tree. Note that last year, Wadkin et al. published a related theoretical paper on the same dataset. The authors start with an intuitively simple ‘base model’ consisting of N=16 cells with OCT4 level and cell cycle duration taken from the corresponding experimental distributions. They test a plethora of model variations with the aim to capture the exact experimental data. In the end, the degree of complexity and the number of parameters go beyond reasonable. In general, this manuscript reads more like a personal learning experience than a research article. This study does not have a high impact, neither is it original. I would recommend rejecting it, as a minimum major revision is needed.

MAJOR POINTS

1. Line 110 and lines 166-174: Since the entire manuscript builds on the finding H=0.38, please show the derivation of the Hurst exponent, i.e., fit on a log-log plot, in the supplement. In the discussion, please compare this value to other studies.

2. Fig 2c: Why is it at all needed to report a synchronized model? That being said, it looks like Fig 2c matches better if all cells have undergone exactly half their cell cycle at the start of the model. Shifting the blue curve six hours to the left makes a synchronized and non-synchronized look qualitatively similar, making it harder to say that the non-synchronized is the better of the two.

3. Line 191-211: Both absorbing and reflecting boundary conditions introduce two new parameters that are not mentioned nor discussed. The lower boundary is reasonably set at 0. However, the higher boundary is ill-defined, and the results in Fig S2c-f depend on this parameter. In general, the arguments for introducing absorbing and reflecting boundary conditions are weak.

4. Line 248: Please, describe what is r? What does it represent? Please elaborate on how this parameter is estimated. Specifically, in line 248, r=0.02 is used, vs. in line 265, r=0.005 is used.

5. After line 279: What is additive and multiplicative noise in this biological system? Specify and exemplify. Why is it important to consider both types of noise in this case? Please elaborate on why additive noise might be dominant at early times and multiplicative at late times. What occurs to the cells when they transition from additive to multiplicative noise?

6. Line 283: The time point 20h seems completely arbitrarily chosen. The same applies to the 25h in line 303. What scientific biological arguments underlie these time points?

7. Table 1: The number of parameters has now grown beyond 10. Overall, the authors need to use solid scientific arguments for evaluating whether a model captures the experiments (rather than saying it “is qualitatively similar,” line 269) or is better than another model (rather than saying it “is an improvement,” line 272). Occam’s razor and “how to fit an elephant” become relevant. This comment applies to the rest of the manuscript.

8. Line 323: While this is part of the experimental dataset and it is a motivation for this study, it takes up very little space in the manuscript, and it blows up the number of parameters to approximately 20. Please streamline the story and minimize the number of parameters. Please distinguish between those parameters that are free vs. those that are constrained, and present a parameter scan of the free parameters.

9. Line 526: Can it be true that only one study reports OCT4 expression in hESC? To the best of my knowledge, there are other published datasets available that at least deserve to be mentioned, especially If the authors attempt to develop a general mathematical model.

10. The discussion reads more like an extended summary than an actual scientific discussion.

11. Please explain why a Gillespie model is never applied?

12. Please, make the MATLAB scripts generating the models and the figures fully available.

MINOR POINTS

1. Abstract line 3: known is misspelled.

2. The abstract is lacking a conclusion.

3. Fig 1a: Why is this figure not the same as Fig 2a in Ref18?

4. Fig 1b-d: What do the authors want to show by including these panels? Please comment on these in the main text or remove them from the manuscript if they are not strictly needed to tell the story.

5. Fig 2a-b: It is easier to compare the orange curve and the blue bars by plotting CDFs and not PDFs.

6. Fig 2b: Does this distribution change with time or cell state?

7. 30 cells in line 79 contra 16 cells in line 188? Please clarify the distinction.

8. Line 83: Position information is not essential for this study. There is no need to state it.

9. Line 87: It might be more meaningful to state the mean and standard deviation cell cycle duration.

10. Line 92: What defines the unknown category? Please be more specific about how the pluripotent and differentiated states are defined. If specific thresholds are used, then mention them.

11. Line 121: What is the number of cells in each colored-curve in Fig 1c? Please provide those numbers and a statistical test on the statement in this line number.

12. Line 134: Here, please restate the purpose of developing these models.

13. Line 152: What is the required number of division events for OCT4 cells? Please state it.

14. Line 180: Why is the built-in MATLAB function not used over the homemade function?

15. Lines 191 and 194: This should be Fig S2a specifically. Also, be more specific in lines 205 and 211.

16. Line 206-210: These sentences are hard to read. Please, rephrase.

17. Line 450-452: Why are chemical reactions additive noise? Please, explain.

18. Ref42 is a webpage. When was it last updated, and when was it accessed?

Reviewer #2: Oct4 is a key regulator of pluripotency. In this work, the authors explore stochastic models that can describe experimental data on Oct4. The study is based upon a previous publication where Oct4 dynamics in a colony of human embryonic stem cells were followed by time-lapse fluorescence microscopy, prior to and after the addition of the differentiation-inducing signal BMP4. Based upon the expression of the marker Cdx2, cells are classified as either pro-pluripotent or pro-differentiated over the whole lineage. A range of increasingly complex phenomenological stochastic models are presented, and their ability to describe key features of the data is examined. All the models are based upon simulating oct4 dynamics over a cell cycle, independently of cell cycle length, which is chosen from the experimental distribution of cell cycle durations. Ultimately, the authors show that a model based upon a logistic equation with a time-dependent Allee effect, additive and multiplicative noise, can reproduce almost all the key features of the data.

The paper is very clear, the mathematical assumptions well explained and the conclusions are critically presented.

My only minor comments are:

- Although the logistic equation is found to be an effective phenomenological equation to reproduce key features of the data, it deviates from the common reasoning about pluripotency factors as regulating each other with feedback interactions. I would encourage the authors to discuss a bit more what other model forms could be used to account for some of the biological knowledge known, and whether maybe this would simplify the noise terms. In particular, I wonder if the antipersistence modelled by the fBm noise could also emerge as a result of negative feedback, where increases in oct4 are followed by decreases, and vice-versa, and whether in this case simpler noise models may be enough. If there is any prior work pointing to this it would be interesting to highlight.

- in ref. 33, a prior publication on the same data by the same authors, and in the original experimental publication, the authors report that BMP4 is added at t=43h, whereas in the present work it says treatment is added at t=40h (line 80, line 99). This should be fixed or the discrepancy clarified.

I was not able to find the data/code in the data.ncl so I suppose this will be done prior to publication.

6. PLOS authors have the option to publish the peer review history of their article (what does this mean?). If published, this will include your full peer review and any attached files.

Reviewer #1: No

Reviewer #2: **Yes: **Rosa Martinez Corral

---

## [Author Response · Author response to Decision Letter 0]

4 May 2021

We attach a pdf file containing our point by point responses to the reviewer comments.

---

## [Decision Letter · Decision Letter 1]

26 May 2021

PONE-D-21-04839R1

A mathematical modelling framework for the regulation of intra-cellular OCT4 in human pluripotent stem cells

PLOS ONE

Dear Dr. Wadkin,

Thank you for submitting your manuscript to PLOS ONE. After evaluating your revised manuscript, reviewer 1 still has outstanding issues that should be addressed prior to publication. If you decide to revise your manuscript according to these comments, the resubmitted version would be sent again to that reviewer for their final assessment.

We look forward to receiving your revised manuscript.

Kind regards,

Jordi Garcia-Ojalvo

Academic Editor

PLOS ONE

Reviewers' comments:

Reviewer's Responses to Questions

**Comments to the Author**

1. If the authors have adequately addressed your comments raised in a previous round of review and you feel that this manuscript is now acceptable for publication, you may indicate that here to bypass the “Comments to the Author” section, enter your conflict of interest statement in the “Confidential to Editor” section, and submit your "Accept" recommendation.

Reviewer #1: (No Response)

Reviewer #2: All comments have been addressed

2. Is the manuscript technically sound, and do the data support the conclusions?

Reviewer #1: Partly

Reviewer #2: Yes

3. Has the statistical analysis been performed appropriately and rigorously? 

Reviewer #1: N/A

Reviewer #2: Yes

4. Have the authors made all data underlying the findings in their manuscript fully available?

Reviewer #1: Yes

Reviewer #2: Yes

5. Is the manuscript presented in an intelligible fashion and written in standard English?

Reviewer #1: Yes

Reviewer #2: Yes

6. Review Comments to the Author

Reviewer #1: SUMMARY

Last time I argued for rejecting this manuscript because too many changes would be needed to make it publishable. I did not find the manuscript of high impact nor - and more critical to PLOS One - original. Since then, the authors have improved the manuscript. However, major changes are still required. In particular, to make this study publishable in PLOS One, it must be valid. I write comments related to the validity as major points below with note that these are not time-consuming but critical for publication in PLOS One. Other remarks of less importance to PLOS One are provided as minor points.

MAJOR POINTS

(1) Abstract + lines 4-5: ‘drug discovery’ is not a clinical application, and ‘personalized medicine’ is overshooting the perspectives of hPSCs and the long-term perspectives of the manuscript. Sentences like these oversell the manuscript. Please down tune. The abstracts of references 1-6 do not even mention personalized medicine.

(2) I cannot run the scripts to reproduce Figs 1-2 due to an unrecognized field name, ‘Fate.’ Please provide the cell_data struct directly in the repository with all required field names. Also, I cannot reproduce Fig 6 among others due to an unknown function or variable ‘shadedErrorBar’. Please fix all these errors.

(3) Unsatisfactory, the authors still do not present a log-log H curve. Given what the authors present, I am unsure whether the authors exclude any cells with few time points from the H estimation. The shortest time series of a cell in the dataset is 0.25h = 3 time points. Do the authors also calculate an H value for them? The genhurst command recommends at least 50-time points for reliable H estimation.

(4) Lines 90-93: Please show this in the supplement or refer to it if it’s already published.

(5) Line 488: I would not call these “statistical models,” conceptual or descriptive maybe, but not statistical. No statistical tests have been applied, and the models have not been used in any quantitative way.

(6) Lines 548: I cannot see how ref 68 backs up this statement. Please elaborate or remove the statement along with the reference.

(7) Figs 1a and 3a: Please keep colors consistent across (all) figures. Is it correct that the orange lines in Fig 3a are identical to the blue lines in Fig 1a? And if so, why is there an orange line above 2500 at 39h in Fig 3a and no blue line at the same intensity and time in Fig 1a?

(8) Fig 5 caption: “For pro-pluripotent cells, the carrying capacity reduces at 20 hours, whilst the carrying capacity for pro-differentiated cells is constant.” If I am right, this does not match the information provided in Table 2. In Table 2, K is reduced at 25h for both cell types!

(9) Figs S7-S8 and Eq. 6: I am getting confused in the notation. What is N? Relative to O? What is the value for N in Fig S7? And what is the value for O in Fig S8? Both refer to Eq. 6 that contains N and O, but N is never introduced in the main text.

(10) Tables 2-3: For differentiated cells, K=1000 after 25h (and <43h) in Table 2, whereas in Table 3, K=1100 after 25h (and <43h) for differentiated cells as well. Is this difference in the K value made on purpose? And why? No comment is provided on this in the main text.

(11) Tables 2-3: Changing sigma A and sigma M from Table 2 to Table 3 makes it difficult to compare Fig 5 with Fig 6. What is the argument that these parameters that affect the simulations <43h should change when introducing an experimental change at 43h? Please comment on this change in the main text or be consistent with the choice of parameter values.

MINOR POINTS

(1) Abstract: What do the authors mean by “important aspects” and “distinct advantages”? Please specify. Also, the use of “sophisticated” seems odd.

(2) Line 86: “Denote the times between(?) cell birth and division”

(3) Line 88: Please remove additional “.”

(4) Line 128: Please list the supplementary figures in the order they are mentioned. Fig S6 is mentioned before Fig S2.

(5) Line 132: change “shown” to “show” and line 133: change “shown” to “as shown.”

(6) Line 163: To increase reproducibility, the authors could consider stating the exact number of division events used as requested already during the last round of peer-review, i.e., 200 for diff cells and 500 for pp cells, etc.

(7) Lines 297-298: Please elaborate on the statement in these two lines.

(8) Line 302: What do the authors mean by “other cellular components”?

(9) Lines 512-514: This is also imprecise and vague. Please either eliminate or elaborate on what “variety of experimental and imaging conditions” the authors mean.

(10) Lines 601-606: These lines are based on my previous comment. However, I am not satisfied with the authors’ reply. Although this is not strictly needed to make the manuscript publishable in PLOS One, I would appreciate, if the authors early on in the manuscript introduced a few success criteria that the models need to satisfy to make them ‘good’ models. The authors need a reliable way to compare the models, whether this is a Bayesian, statistical, or descriptive comparison. In Table 4, the authors give it a try. However, these ‘features’ are not presented early on in the manuscript nor clearly in Table 4.

(11) Fig 1a-b: Please add legends to (all) figures to increase readability, especially for Fig 1a-b.

(12) Fig 3b caption: Please write “estimated carrying capacity (vertical line, K = 1290)”

(13) Figs S6 and S9: Why are the authors presenting the mode? What information does the mode add that the median does not? Often the authors refer only to panel S6c but not S6a; why? Actually, in line 373, the mean and median are mentioned in parenthesis, but S6 shows the mode and the median.

Reviewer #2: All the comments have been addressed. Just be aware of a couple of typos, in line 513 and line 515-516.

7. PLOS authors have the option to publish the peer review history of their article (what does this mean?). If published, this will include your full peer review and any attached files.

Reviewer #1: No

Reviewer #2: **Yes: **Rosa Martinez-Corral

---

## [Author Response · Author response to Decision Letter 1]

5 Jul 2021

We have attached our detailed response as a file. Many thanks to the reviewers once again.

---

## [Decision Letter · Decision Letter 2]

8 Jul 2021

A mathematical modelling framework for the regulation of intra-cellular OCT4 in human pluripotent stem cells

PONE-D-21-04839R2

Dear Dr. Wadkin,

We’re pleased to inform you that your manuscript has been judged scientifically suitable for publication and will be formally accepted for publication once it meets all outstanding technical requirements.

Kind regards,

Jordi Garcia-Ojalvo

Academic Editor

PLOS ONE

---

## [Editor Report · Acceptance letter]

26 Jul 2021

PONE-D-21-04839R2 

A mathematical modelling framework for the regulation of intra-cellular OCT4 in human pluripotent stem cells 

Dear Dr. Wadkin:

I'm pleased to inform you that your manuscript has been deemed suitable for publication in PLOS ONE. Congratulations! Your manuscript is now with our production department. 

Kind regards, 

on behalf of

Dr. Jordi Garcia-Ojalvo 

Academic Editor

PLOS ONE